# Mechanisms of Oncogenesis by HTLV-1 Tax

**DOI:** 10.3390/pathogens9070543

**Published:** 2020-07-07

**Authors:** Suchitra Mohanty, Edward W. Harhaj

**Affiliations:** Department of Microbiology and Immunology, Penn State College of Medicine, Hershey, PA 17033, USA; smohanty@pennstatehealth.psu.edu

**Keywords:** HTLV-1, Tax, NF-κB, ATLL, apoptosis

## Abstract

The human T-cell lymphotropic virus type 1 (HTLV-1) is the etiological agent of adult T-cell leukemia/lymphoma (ATLL), a neoplasm of CD4+CD25+ T cells that occurs in 2–5% of infected individuals after decades of asymptomatic latent infection. Multiple HTLV-1-encoded regulatory proteins, including Tax and HTLV-1 basic leucine zipper factor (HBZ), play key roles in viral persistence and latency. The HTLV-1 Tax oncoprotein interacts with a plethora of host cellular proteins to regulate viral gene expression and also promote the aberrant activation of signaling pathways such as NF-κB to drive clonal proliferation and survival of T cells bearing the HTLV-1 provirus. Tax undergoes various post-translational modifications such as phosphorylation and ubiquitination that regulate its function and subcellular localization. Tax shuttles in different subcellular compartments for the activation of anti-apoptotic genes and deregulates the cell cycle with the induction of DNA damage for the accumulation of genomic instability that can result in cellular immortalization and malignant transformation. However, Tax is highly immunogenic and therefore HTLV-1 has evolved numerous strategies to tightly regulate Tax expression while maintaining the pool of anti-apoptotic genes through HBZ. In this review, we summarize the key findings on the oncogenic mechanisms used by Tax that set the stage for the development of ATLL, and the strategies used by HTLV-1 to tightly regulate Tax expression for immune evasion and viral persistence.

## 1. Introduction

The human T-cell lymphotropic virus type 1 (HTLV-1) is the first oncogenic retrovirus discovered in humans in the early 1980s by two independent research groups in Japan and America [1,2]. HTLV-1 is an enveloped complex retrovirus that belongs to the family *Retroviridae* and genus deltaretrovirus. This genus also includes three additional HTLV members, HTLV-2, -3, and -4 [3,4,5]. HTLV-1 is the causative agent of a neoplasm of CD4+CD25+ T cells known as adult T-cell leukemia/lymphoma (ATLL), which consists of four clinical subtypes: smoldering, chronic, lymphoma, and acute. Whereas smoldering and chronic ATLL represent more indolent forms of the disease, lymphoma and acute ATLL are highly aggressive with a dismal prognosis and a median survival of ~6 months. HTLV-1 infection is also associated with a variety of inflammatory and autoimmune diseases such as HTLV-1-associated myelopathy/tropical spastic paraparesis (HAM/TSP), uveitis, arthritis, dermatitis and bronchiectasis as a result of an immune-deficient state [6,7]. Although HTLV-1 and HTLV-2 share a similar genomic structure with 70% nucleotide similarity, no clear correlation between HTLV-2 and lymphoproliferative disease has been established [8].

HTLV-1 transmission occurs by three different routes: vertical transmission from a carrier mother to her infant through breast-feeding, horizontal transmission of infected lymphocytes through sexual contact, and parenteral (e.g., intravenous drug injection, blood transfusion and organ transplant) [9,10,11,12]. As vertical transmission is the primary route of infection to new individuals, HTLV-1 infection groups are mostly clustered in specific geographical areas of the world including southern Japan, sub-Saharan Africa, the Caribbean basin, South America (in particular Brazil, Colombia, Chile, and Peru), parts of the Middle East (Iran), and Australia [13]. Recent epidemiological studies have revealed extremely high (>40% of adults) HTLV-1 infection rates in indigenous communities in central Australia (e.g., Alice Springs) [14]. However, epidemiological data are very limited in highly populated regions, such as China and India, therefore the number of infected people world-wide may be underestimated. Approximately 10 million HTLV-1-infected people world-wide remain asymptomatic throughout life; however, 2–5% of infected individuals develop aggressive ATLL 40–60 years after infection. Thus, the majority of ATLL patients contracted HTLV-1 at birth from a carrier mother rather than adulthood infection. However, viral RNA is rarely detected in the plasma of infected individuals as HTLV-1 persists for decades in the host by cell-to-cell transmission of viral particles (*de novo* infection) and clonal proliferation or mitotic expansion of infected cells while limiting its replication [15,16,17]. ATLL has a poor prognosis and survival which is influenced by key prognostic factors such as poor performance status, elevated LDH levels, a minimum of four involved lesions, hypercalcemia, minimum age of 40 years, thrombocytopenia, eosinophilia, bone marrow involvement, high interleukin-5 serum levels, C–C chemokine receptor 4 (CCR4) expression, lung resistance-related protein, p53 mutation and p16 deletion [18]. The current treatment options for ATLL include watchful waiting, zidovudine plus interferon-alpha (AZT/IFN), multi-agent chemotherapy or allogeneic hematopoietic stem cell transplantation (allo-HSCT); however, chemoresistance prevents long-term disease-free survival. HTLV-1 is a highly oncogenic virus that manipulates host cellular signaling pathways to induce the hallmarks of cancer with successful evasion of immune-surveillance. The oncogenic ability of HTLV-1 is mediated by viral gene products and their interaction with host proteins to alter their function and thus favor viral infection and persistence. In this review, we will discuss recent findings on the mechanisms of HTLV-1-mediated transformation of T lymphocytes by the viral oncoprotein Tax. 

## 2. HTLV-1 Genomic Structure and Modes of Entry

The HTLV-1 virion is enveloped, ~100 nm in diameter, and carries two identical strands of genomic RNA inside a protein capsid. The viral capsid also contains functional protease (pro), integrase (IN), and reverse transcriptase (RT) enzymes. CD4^+^ T lymphocytes harbor the vast majority (>90%) of the HTLV-1 viral load in vivo [19]. However, HTLV-1 can also infect CD8+ T cells, B cells, monocytes and dendritic cells (DCs) as additional reservoirs of infection [20,21]. HTLV-1 cell-free virions are less infectious in most cell types, except for DCs [22], and cell-to-cell contact for transmission of the virus can occur through a variety of mechanisms, including a virological synapse, viral biofilm, cellular conduit or tunnelling nanotube [23,24,25,26,27]. The virological synapse is a virus-induced specialized area of cell–cell contact that promotes transmission of HTLV-1 to an uninfected T cell. Virus-infected T cells form the virological synapse by intercellular adhesion molecule 1 (ICAM-1) binding to its ligand, the integrin lymphocyte function-associated antigen 1 (LFA-1) at the cell surface, followed by reorientation of the microtubule organizing center (MTOC) in HTLV-1-infected T cells towards the site of cell–cell contact in uninfected cells [28,29]. The virological synapse facilitates the accumulation of complexes of HTLV-1 core protein (Gag), and the viral RNA genome at the area of cell–cell contact followed by transfer to the uninfected cell.

HTLV-1 entry into cells depends on sequential interactions with three host components. The HTLV-1 viral envelope (Env) is thought to initially interact with heparin sulfate proteoglycans (HSPGs) followed by the formation of a complex with neuropilin-1 (NRP1) and glucose transporter-1 (GLUT1), which results in fusion [30,31,32,33]. Subsequently, the viral RNA is delivered into the cytoplasm of infected cells and undergoes reverse transcription to form double-stranded DNA for integration into the host genome as a provirus. The HTLV-1 genome is approximately 9 kb in length; however, the viral genome has the potential to express multiple products by exploiting various strategies such as frameshifting, polycistronic translation, alternative mRNA splicing and protease-mediated cleavage of large viral proteins into smaller proteins with specific functions. The viral genome contains flanking regions of 5’ and 3’ long terminal repeats (LTRs), containing *cis*-acting promoter elements that drive the transcription of viral genes such as the structural protein Gag (capsid, nucleocapsid, matrix), Pro (Protease), polymerase (Pol) and Env from unspliced/singly spliced mRNA [34]. The regulatory and accessory proteins of HTLV-1 are generated by alternatively spliced mRNA transcripts [35,36,37,38]. During viral assembly, the viral genomic RNA, along with Gag, Env, and Gag-pol proteins are transported to the plasma membrane of cells to form an immature viral particle [39]. The matrix protein binds directly to the inner leaflet of the cell membrane, and the capsid protein drives Gag–Gag interactions to form the Gag lattice in immature particles as well as the viral capsid core in mature particles [40,41]. The nucleocapsid protein interacts with and coats the viral RNA genome. Upon release from the cell surface, the budding viral particle undergoes maturation triggered by viral-mediated proteolysis to form an infectious viral particle [42]. During virus maturation, Gag is cleaved by the viral protease, after which capsid proteins reassemble into a capsid core that encapsulates the nucleocapsid RNA complexes and the viral replication enzymes. 

The HTLV-1 regulatory genes *Tax*, *Rex*, *p21*, *p12*, *p13* and *p30* are all encoded by various open reading frames (ORFs) in the pX region located in the 3’ end of the genome [43,44,45]. The accessory genes *p12*, *p13* and *p30* play significant roles in establishing and maintaining viral persistence, whereas Rex regulates post-transcriptional viral gene expression and increases the stability of viral RNA for the latency phase of the viral life cycle [46,47,48,49]. The HTLV-1 basic leucine zipper factor (HBZ) regulatory protein is transcribed in an antisense manner directed by the 3’ LTR and is consistently expressed in ATLL [50,51]. HBZ plays a crucial role in the oncogenesis and maintenance of the transformed phenotype [52]. Of note, the 5’ LTR is frequently deleted and methylated, whereas the 3’ LTR remains intact in all cases of ATLL [53]. The HTLV-1 *trans*-activator protein Tax is expressed from a doubly spliced mRNA transcript and is a multifunctional protein that triggers a plethora of events, such as the activation of cell signaling pathways that drive proliferation and survival of T cells that can promote their immortalization and transformation [54,55]. However, a complete understanding of the mechanisms of Tax-mediated modulation of host cellular proteins and oncogenesis is yet to be achieved. 

## 3. Oncogenic Functions and Regulation of Viral Gene Expression by HTLV-1 Tax

The oncogenic function of Tax has been demonstrated *in vivo* by the generation of different Tax transgenic mouse strains, which develop distinct tumors and/or inflammatory lesions depending on the promoter used to drive Tax expression [56,57,58,59]. Tax can transform murine fibroblasts [60], immortalize primary human CD4+ lymphocytes [61] and induce ATLL-like disease in transgenic mice [62]. Moreover, Tax-immortalized lymphoid cells resemble the phenotype of HTLV-1 transformed T cells and possess the ability to induce tumors in immunodeficient mice, which closely resemble ATLL [62,63,64]. Tax-induced changes in gene expression and epigenetic modifications persist in ATLL, despite the common downregulation of Tax [65]. Tax profoundly alters gene expression and dysregulates genes involved in cell cycle regulation, cell survival, and cell proliferation, as well as cytokines and cytokine receptors [66], and these genes are also dysregulated in ATLL [67]. Similar to ATLL cells, Tax immortalized cells accumulate H3 lysine 27 trimethylation (H3K27me3) marks, responsible for chromatin condensation and gene silencing and can indirectly suppress diverse target genes through modulation of host epigenetic machinery via interaction with enhancer of zeste homolog 2 (EZH2), the active core subunit of polycomb repressive complex 2 (PRC2) [68].

Tax is a critical regulator of viral gene expression and recruits host transcription factors such as cyclic AMP (cAMP)-response element-binding protein (CREB) and the coactivators, CREB binding protein (CBP) and p300 to the LTR for *trans*-activation of the viral promoter [69]. Tax forms a homodimer and interacts with CREB homodimers or CREB/ATF (CREB-Activating transcription factor family protein) heterodimers to enhance DNA binding affinity for viral TREs (Tax responsive elements) and ensure optimal transcription even in the absence of CREB phosphorylation [70,71]. Interestingly, recruitment of the CBP/300 complex by Tax promotes chromatin remodeling for maximal transcription [72,73], and the addition of Tax to the SWI/SNF (switch/sucrose non-Fermentable) complex could boost the rate of chromatin remodeling of the nucleosome to aid the function of CBP/300 [74]. Moreover, the transcription factor Myocyte Enhancer Factor (MEF)-2 is highly expressed in ATLL and is associated with the stabilization of the Tax/CREB complex to support LTR activation and viral replication [75]. Tax also regulates the binding of repressors to the HTLV-1 promoter via interaction with histone deacetylase 1 (HDAC1) and inhibits binding of HDAC1 to the HTLV-1 promoter [76]. Furthermore, Tax interacts with the histone methyltransferase SUV39H1 for tethering to the LTR to self-limit HTLV-1 viral gene expression in a negative feedback loop [77]. Together, Tax exerts tight control over viral gene expression through selective protein–protein interactions.

Tax controls virus replication and persistence by regulating its localization in different compartments of cells, including the cytosol, nucleus, Golgi apparatus, and endoplasmic reticulum (ER) via different sequences including a nuclear localization sequence (NLS) and nuclear export sequence (NES) [78,79,80,81,82]. However, Tax localization is dynamic and its relocalization to the nucleus can be triggered by stimuli such as genotoxic stress, whereas cell-free Tax secreted into the extracellular space contributes to inflammation and pathogenesis [83,84]. The oncogenic properties of Tax are tightly associated with its ability to dysregulate, manipulate, and exploit host cellular signaling pathways through interaction with multiple cellular factors [85].

## 4. HTLV-1 Tax Activation of NF-κB

NF-κB is an evolutionarily conserved transcription factor family with diverse physiological roles including cell proliferation, apoptosis, oncogenesis, development and the innate and adaptive immune response. NF-κB family members consist of p65 (RelA), c-Rel, RelB, p50/p105 and p52/p100 [86]. NF-κB dimers are sequestered in the cytoplasm as inactive forms bound to a member of the IκB family. IκB proteins can mask the NLS of the NF-κB proteins resulting in their sequestration as latent transcription factors in the cytoplasm [87]. Upon activation, IκB is phosphorylated by the IκB kinase (IKK) complex consisting of the catalytic subunits IKKα and IKKβ, and the regulatory subunit IKKγ (also known as NF-κB essential modulator (NEMO)) [88]. IKK-mediated phosphorylation of IκBs results in their ubiquitination and proteasomal degradation, thus allowing NF-κB to translocate to the nucleus. This NF-κB activation pathway is known as the classical or canonical pathway. There is also a noncanonical NF-κB pathway centered upon the kinases NF-κB-inducing kinase (NIK) and IKKα which trigger the processing of NF-κB2/p100 to p52 [89,90]. RelB/p52 heterodimers can activate a unique subset of genes encoding chemokines and anti-apoptotic proteins. The noncanonical NF-κB pathway is activated by tumor necrosis factor receptor (TNFR) superfamily members such as B-cell activating factor (BAFF), CD40 and lymphotoxin beta receptor (LT-βR). 

NF-κB signaling is normally transiently activated by acute stimuli and terminated as a result of induction by negative feedback inhibitors. However, oncogenic viruses have evolved various strategies to manipulate NF-κB signaling to establish long-term persistence within their hosts, and inhibition of NF-κB signaling impairs the proliferation and survival of virus-infected cells [91,92,93]. Canonical and noncanonical NF-κB pathways are constitutively activated in HTLV-1-transformed cell lines, Tax-expressing cells, and ATLL patient samples, with aberrant expression of cytokines and cytokine receptors [94,95,96,97]. Indeed, Tax acts on multiple levels to induce and maintain pathological NF-κB activation through post-translational modifications (PTMs) (Figure 1). With regard to canonical NF-κB activation, Tax triggers the persistent phosphorylation, ubiquitination, and subsequent proteasomal degradation of IκBα, leading to aberrant NF-κB activation. Tax induces the constitutive phosphorylation and activation of the IKK complex to sustain NF-κB activation [98]. Tax binds to the IKK complex via direct interaction with NEMO to facilitate the recruitment of the catalytic subunits of IKK to activate downstream events [99,100]. Tax may also promote NF-κB activation through dysregulation of IκB proteins since Tax can directly interact with IκB proteins, specifically IκBα [101] and p105 [102], via their ankyrin repeat domains. This interaction promotes NF-κB activation, either by disrupting NF-κB/IκB complexes or by recruiting IκB proteins directly to the proteasome for degradation. Tax can target p105 to the proteasome to accelerate its cleavage to the active form p50 [103]. The precursor NF-κB2/p100 is another target of Tax, as well as IκBβ [104,105]. Tax can also directly engage the NF-κB proteins p50, p52, RelA and c-Rel [106] by binding to the Rel homology domain to trigger their homo- or hetero-dimerization, resulting in an enhancement of DNA binding and transcriptional activity. It has also been reported that Tax, RelA, p50, RNA polymerase II and CBP/p300 co-localize in transcriptionally active nuclear foci [107]. The transcriptional co-activators CBP/p300 are recruited to RelA by Tax to potentiate NF-κB transcriptional activity [108]. 

Tax also chronically activates the noncanonical NF-κB pathway, which contributes to cell survival and oncogenesis. Surprisingly, in contrast to TNFR superfamily members which require only IKKα, Tax requires both IKKα and NEMO to activate the noncanonical NF-κB pathway via formation of a complex that also contains p100 [109]. Tax physically recruits IKKα to p100, triggering its phosphorylation, ubiquitination, and proteasomal processing to p52 [110]. Although NIK appears to be dispensable for Tax-induced noncanonical NF-κB activation [109], overexpression of NIK due to epigenetic changes can drive oncogenic NF-κB activation in ATLL [111].

Tax interacts with a host of signaling proteins to impart persistent NF-κB activation. Tax has been reported to interact with and activate the kinase activity of TGF-β-activating kinase 1 (TAK1) through TAB2 (TAK1 binding protein 2) [112]. Tax also mediates the recruitment of the IKK complex to TAK1 [113]. In addition to activating kinase complexes, Tax can also modulate the activity of ubiquitin E3 ligases such as Ring Finger Protein 8 (RNF8). Tax stimulates RNF8 activity to assemble unanchored long lysine 63 (K63)-linked polyubiquitin chains, which induce IKK and canonical NF-κB activation along with downstream kinases including c-Jun N-terminal kinase (JNK) [114]. Tax also interacts with and recruits the linear (M1-linked) ubiquitin E3 ligase complex LUBAC (linear ubiquitin assembly complex) to NEMO and the IKK complex in HTLV-1-transformed T cell lines, and together with a K63-Ub-specific E3 ligase, generate K63/M1-linked hybrid polyubiquitin chains [115]. Finally, Tax can activate the K63-Ub specific E3 ligase, TRAF6, through direct interaction with a TRAF6-binding motif in its carboxyl-terminus, which stabilizes anti-apoptotic MCL-1 through K63-linked polyubiquitination to inhibit apoptosis [116].

Tax relies on a number of mechanisms and binding proteins to ensure persistent NF-κB signaling including cross-talk between canonical and non-canonical pathways, the establishment of feed-forward signaling loops and inactivation of negative regulators. Tax-mediated activation of the noncanonical pathway suppresses the expression of the WWOX (WW domain containing oxidoreductase) tumor suppressor gene, which inhibits canonical NF-κB signaling [117]. Tax-IKK activation induces the expression of the receptor IL-17RB which promotes a feed-forward signaling loop that sustains canonical NF-κB activation in HTLV-1-transformed T cells [118]. Peptidylproline cis-trans isomerase (PIN1), overexpressed in HTLV-1-infected T-cells and Tax transfected cells, may also contribute to NF-κB activation by Tax [119]. The histone methyltransferase, SMYD3 (SET and MYND domain containing 3) has been shown to bind to Tax and promote NF-κB activation, and SMYD3 also regulates the nucleo-cytoplasmic shuttling of Tax [120]. As part of its mechanism to induce chronic NF-κB signaling, Tax targets and inactivates several negative regulators of NF-κB signaling. At the level of IKK activation, Tax can inhibit the serine/threonine protein phosphatase 2A (PP2A) which dephosphorylates and inactivates IKK [121]. A20 (also known as TNFAIP3) is a ubiquitin-editing enzyme and negative regulator of NF-κB activation that can form a complex with the selective autophagy receptor TAX1BP1 and the E3 ligases Itch and RNF11 by IKKα-mediated phosphorylation of TAX1BP1; this complex inhibits NF-κB signaling to maintain transient NF-κB activation [122,123]. However, Tax interacts with TAX1BP1 and inhibits IKKα-induced phosphorylation of TAX1BP1 to disrupt its interactions with A20 and Itch, thereby inactivating a critical checkpoint for NF-κB inhibition [124]. A20 can also disrupt interactions between E2 ubiquitin conjugating enzymes (e.g., Ubc13) and E3 ubiquitin ligases to inhibit NF-κB signaling; however, Tax can block A20-Ubc13 binding to sustain Ubc13-dependent K63-linked polyubiquitination of substrates [125]. In addition to TAX1BP1, Tax can also interact with other autophagy receptors to promote NF-κB activation. NEMO-related protein (NRP), also known as Optineurin, is a selective autophagy receptor that has been demonstrated to interact with ubiquitinated Tax through its ubiquitin-binding domain (UBD), for its stabilization and to enhance NF-κB activation by Tax [126]. A recent study also demonstrated that Tax potentiates NF-κB activation through its interaction with the selective autophagy receptor SQSTM-1/p62 [127]. Furthermore, Tax also interacts with the cytoplasmic tail of cell adhesion molecule 1 (CADM1), and this interaction facilitates Tax interaction with NEMO, TAX1BP1 and NRP for activation of the IKK complex [128]. CADM1 also supports the K63-linked polyubiquitination of Tax by Ubc13 for the activation of NEMO [128]. The deubiquitinase cylindromatosis (CYLD) can interact with and inhibit Tax ubiquitination and downstream IKK activation; however, CYLD is constitutively phosphorylated and inactivated in HTLV-1-transformed cells [129]. Taken together, Tax interacts with an array of proteins and utilizes diverse strategies to promote persistent NF-κB signaling.

## 5. Post-translational Modifications (PTMs) of Tax and Its Contribution to Cellular Transformation

Like other viruses, HTLV-1 hijacks the host cellular machinery to promote PTMs of viral, as well as cellular proteins to enhance viral replication and persistence, and immune evasion. These PTMs can also influence signaling pathways contributing to uncontrolled cell proliferation and transformation. Tax PTMs regulate its cellular localization, *trans*-activation, and interaction with host proteins to modulate their function in order to enhance viral persistence and spread. Tax is subject to multiple PTMs including ubiquitination, SUMOylation, acetylation and phosphorylation, which influence Tax protein-protein interactions and trafficking to different compartments of the cell, and ultimately modulate distinct Tax functions. It is likely there is cross-talk or interdependence among the different PTMs of Tax; however, the chronological order of these modifications essential for NF-κB activation and the enzymes involved in the different modifications are not fully realized. 

### 5.1. Tax Ubiquitination

Mutagenesis studies have revealed that Tax undergoes ubiquitination at multiple lysines including K263, K280 and K284, which modulates its stability, cellular trafficking and NF-κB signaling [81,130]. Although ubiquitinated Tax binds to the proteasome, this does not induce its degradation suggesting that Tax ubiquitination has non-proteolytic functions [130,131]. Polyubiquitin chains can be linked either head-to-tail (known as linear or M1) or via one of seven lysine residues (K6, K11, K27, K29, K33, K48 and K63). The K63-linked polyubiquitination of Tax supports NF-κB activation, likely by acting as a molecular scaffold to recruit kinases such as IKK and TAK1 [112,132]. Tax ubiquitination also promotes the relocalization of NEMO and the IKK complex to a perinuclear region colocalizing with markers of the Golgi or centrosome [133,134]. The E2 ubiquitin conjugating enzyme Ubc13 is essential for K63-linked Tax polyubiquitination and NF-κB activation [132], while an E3 enzyme necessary for Tax K63-linked polyubiquitination has yet to be identified. The ubiquitinated form of Tax interacts with NRP/Optineurin and forms a complex with NEMO for sustained NF-κB activation [126]. Tax is also monoubiquitinated upon genotoxic or cellular stress at K280 and K284 which triggers its nuclear export in a CRM-1-dependent manner [135]. Furthermore, Tax can also be conjugated with K48-linked polyubiquitin chains by the E3 ligase PDLIM2 (PDZ-LIM domain-containing protein 2) and degraded in the nuclear matrix [136]. Tax interacts directly with PDLIM2 through its α-helix motif located between amino acids 236 and 254, and inhibits the NF-κB transcriptional activity of Tax [137]. Interestingly, the expression of PDLIM2 is suppressed by DNA methylation in HTLV-1-transformed T lymphocytes [138]. Tax interacts with the heat shock protein HSP90 to enhance its stability and evade proteasomal degradation in the nuclear matrix [139,140]. Although the vast majority of Tax polyubiquitination is thought to consist of K48- and K63-linked polyubiquitination chains, it remains unknown if Tax can be conjugated with other polyubiquitin chains. Deubiquitinases (DUBs) such as USP20 and CYLD have the potential to deubiquitinate Tax and thus suppress NF-κB activation [129,141]. However, USP20 expression is downregulated, and CYLD deubiquitinase activity is suppressed by phosphorylation in HTLV-1 transformed cell lines [129,141].

### 5.2. Tax SUMOylation

SUMOylation refers to the reversible covalent attachment of a SUMO (small ubiquitin-like modifier) moiety to target proteins executed by sequential enzymatic reactions catalyzed by E1, E2 and E3 enzymes, analogous to ubiquitination. Tax has been shown to be SUMOylated at K280 and K284, sites which can also be ubiquitinated [142]. The modifications of Tax through ubiquitination and SUMOylation are thought to regulate its shuttling from the cytoplasm to the nucleus where it interacts with p300, RelA and NEMO exclusively in nuclear bodies [85,143]. The SUMOylated Tax is found in Tax nuclear bodies while ubiquitinated forms of Tax are largely localized in a perinuclear region in Golgi-associated structures [133,142]. The SUMOylation of Tax is mediated by the E2 SUMO-conjugating enzyme, Ubc9, and the inhibition of its catalytic activity impairs Tax conjugation to endogenous SUMO-1 or SUMO-2/3 [144]; however, other host enzymes regulating Tax SUMOylation are yet to be discovered. Although Tax SUMOylation was initially proposed to mediate NF-κB *trans*-activation in the nucleus [145], another study showed that non-SUMOylated Tax protein remains fully functional for activation of the NF-κB pathway [144].

### 5.3. Tax Phosphorylation

Tax is a phosphoprotein which can be phosphorylated at serines 300 and 301 (S300 and S301). Tax phosphorylation promotes its localization to nuclear bodies and activation of gene expression by the ATF/CREB and NF-κB pathways [146]. Tax has also been shown to be phosphorylated at additional sites including T48, T184, T215 and S336; of these phosphorylation sites only T48 and T215 were implicated in the regulation of either CREB and/or NF-κB [147]. It remains unknown what kinases can phosphorylate Tax and where in the cell Tax phosphorylation occurs.

### 5.4. Tax Acetylation

Tax has also been shown to be acetylated at lysine 346 (K346) and phosphorylation of Tax may serve as a prerequisite event for its acetylation [148]. Tax acetylation likely occurs in the nucleus where acetylated Tax interacts with p300 in nuclear bodies. The functional consequence of Tax acetylation is mostly correlated with NF-κB activation. Tax acetylation is mediated by p300 and is counteracted by HDAC7 in nuclear bodies; notably a Tax acetylation mutant (K346R) is impaired in the transformation of Rat-1 cells [149].

## 6. HTLV-1-mediated Inhibition of Apoptosis

In cancer cells, there is a loss of balance between cell division and cell death due to dysregulated signaling pathways. Programmed cell death or apoptosis is tightly regulated by pro- and anti-apoptotic factors, and evasion of apoptosis can lead to the accumulation of genetic abnormalities and malignant transformation. Like other oncogenic viruses, HTLV-1 specifically targets apoptosis signaling mechanisms in order to suppress apoptosis for the uncontrolled proliferation of virally-infected cells which can eventually result in tumorigenesis. The evasion of apoptosis by cancer cells can be broadly categorized into two groups: (1) an imbalance between pro- and anti-apoptotic proteins and (2) impaired caspase signaling and compromised death receptor signaling (Figure 2).

### 6.1. The Imbalance between Pro- and Anti-apoptotic Proteins

Tax modulates the expression of numerous genes regulating cell survival [150]. For example, Tax enhances the expression of X-linked inhibitor of apoptosis (XIAP) through the activation of NF-κB signaling [151]. The anti-apoptotic protein survivin is overexpressed in HTLV-1-infected T cells as well as in ATLL through Tax-mediated stimulation of NF-κB signaling [152,153,154]. Interestingly, HBZ RNA, which has oncogenic activity, can induce survivin expression by activating its promoter [155]. HTLV-1 drives the expression of anti-apoptotic BCL-2 and Bcl-xL through Tax-mediated activation of transcription factors NF-κB, c-Jun and JunD [156]. Tax also upregulates the expression of the BCL-2 family member Bfl-1/A1 via the NF-κB pathway, and Bfl-1 knockdown diminished the survival of HTLV-1-infected T cell lines [157]. Myeloid cell factor-1 (MCL-1) is a death inhibiting member of the BCL-2 family which is highly expressed in B-cell chronic lymphocytic leukemia (B-CLL), acute myeloid leukemia (AML), and acute lymphoblastic leukemia (ALL) [158]. HTLV-1 Tax maintains MCL-1 protein stability through interaction with TRAF6, and prevents its degradation from genotoxic stress by K63-linked polyubiquitination [116]. Interestingly, HBZ also promotes MCL-1 stabilization by disrupting the interaction between CUL1 and Skp1 in the SCF (Skp, Cullin, F-box) ubiquitin ligase complex [159]. Proapoptotic BCL-2-like protein 11 (BCL2L11; commonly called BIM) is downregulated in ATLL cell lines both at the mRNA level and through BIM protein degradation by Tax; re-expression of BIM induced apoptosis in ATLL cell lines [160]. HBZ has been shown to suppress transcription of the *Bim* gene by inhibiting the function of the FoxO3a transcription factor [161]. Together, it is clear that both Tax and HBZ exhibit overlapping and complementary functions to inhibit cell death.

Tax also targets tumor suppressors to enhance cell survival and sustained cell proliferation. Tax inhibits the tumor suppressor p53 through either CREB or NF-κB pathways to induce immortalization of HTLV-1-infected T cells [162,163]. Tax promotes the constitutive phosphorylation of p53 at serines 15 and 392 (S15 and S392) for functional inactivation [164], and also represses p53-mediated *trans*-activation by CBP/p300 sequestration [165,166]. Since Tax functionally inhibits p53, the *TP53* gene is not mutated in all ATLL patients, but rather ~30% of ATLLs exhibit p53 mutations [167,168]. 

Ras proteins are small GTPases that regulate cell differentiation, proliferation, and survival, and when aberrantly expressed or dysregulated contribute to oncogenesis. Tax targets and induces the expression of Ras family members to protect cells from apoptosis [169]. Tax also targets Ras-Raf-MEK signaling through interaction with Erbin (ErbB2 binding protein) to augment cancer cell proliferation [170]. HTLV-1-infected T cells display an elevated level of cyclic AMP (cAMP), via the Tax protein, for long-term survival through increased metabolism [171]. Interestingly, Tax targets and activates the mTOR signaling pathway and induces phosphorylation of p70S6K and ribosomal protein S6 in T cells for uncontrolled cell growth, even in the absence of the growth factor IL-2 [172].

The survival and proliferation of HTLV-1-infected T cells also depends on Tax-driven dysregulation of cytokines [173]. Tax promotes the expression of multiple cytokines and their cognate receptors including IL-6/IL-6R and IL-2/IL-2Rα to induce cell proliferation and suppress apoptosis [174,175,176,177]. Similarly, IL-9 and IL-13 expression is also induced by Tax in an NF-κB-dependent manner to support the proliferation and survival of infected cells [178,179,180].

### 6.2. Impaired Caspase Signaling and Compromised Death Receptor Signaling

The caspase family of proteases induces apoptosis and are categorized as initiators (i.e., caspase 2, 8, 9 and 10) or effectors (i.e., caspases 3, 6 and 7) based on their function. Tax drives the expression of the caspase inhibitor XIAP through activation of NF-κB signaling, and XIAP inactivates caspases-3, -7 and -9 for the inhibition of apoptosis [151,181,182]. Tax also inactivates caspase-3 through the inhibition of caspase-8 [151]. The inactivated isoform of caspase-8 is also responsible for the inhibition of apoptosis through the attenuation of Fas-mediated signaling in ATLL [183]. CD95/Fas death receptor-mediated apoptosis is inhibited by cFLIP, and Tax protects against CD95-mediated cell death through the induction of cFLIP in HTLV-1-infected cells [184,185]. The stress-responsive gene A20 serves as a dual protector against tumor necrosis factor (TNF)-mediated apoptosis through the inhibition of caspase-8 signaling as well as inflammation through the suppression of NF-κB signaling [186]. Tax promotes the expression of anti-apoptotic A20 to protect from cell death; however, NF-κB remains persistently activated, even in the presence of A20 [187,188].

## 7. HTLV-1 Deregulation of Cell Cycle and DNA Repair

The highly regulated multi-step process of cell cycle regulation and DNA repair safeguards the errorless inheritance of genetic material; however, oncogenic viruses such as HTLV-1 modulate the function of cell cycle regulatory proteins including cyclin/CDK complexes as well as key DNA repair enzymes, resulting in increased error frequency and uncontrolled cell proliferation [189]. Tax induces the hyperphosphorylation of the tumor suppressor Rb through the interaction with cyclin-dependent kinase 4 (CDK4) for the continuous progression through the G1 phase of the cell cycle [190,191]. Tax also interacts with the E2F transcription factor to transcribe E2F-dependent S phase genes to enable entry into the S phase of the cell cycle [192]. Tax also controls the function of cyclin-CDK complexes through the inhibition of CDK4 inhibitors including p15, p16, p18 and p19 to facilitate S phase entry and cell proliferation, and genetic alterations in these genes have been identified in ATLL [193,194,195,196,197,198,199]. Notably, Tax also facilitates the proteasomal degradation of the hypo-phosphorylated form of Rb to stabilize E2F levels [200].

The tumor suppressor protein p53 also regulates G1 to S transition through the expression of cell cycle control proteins such as p21^waf1/cip1^, which binds and inactivates CDK2 to mediate cell cycle arrest in response to various cellular stresses to prevent progression to the S phase of the cell cycle until DNA repair is completed [201]. However, Tax inactivates p53 signaling while upregulating p21^waf1/cip1^ and CDK2 in a p53-independent manner in HTLV-1-infected T cells to sequester p21^waf1/cip1^ from cyclin E/CDK2, possibly as a mechanism for the active cyclin-CDK complex to phosphorylate Rb proteins and accelerate the progression through the G1/S checkpoint [202,203]. Finally, Tax enhances WIP1 (wild-type p53-induced phosphatase 1) phosphatase activity resulting in diminished phosphorylation of histone H2AX [204]. Thus, via WIP1, Tax can bypass the G1/S checkpoint and suppress the DNA damage response (DDR) to allow for S phase entry in the presence of DNA lesions. 

Similarly, S phase or DNA replication is tightly regulated by cyclin A/CDK2 which phosphorylates the pre-replication protein complex to restrict a single round of replication [205]. Interestingly, Tax suppresses the cyclin A promoter through CREB/ATF resulting in abnormal DNA replication and the accumulation of DNA double-strand breaks (DSBs) that accelerate genomic instability, in part by production of nitric oxide and reactive oxygen species (ROS) [206,207]. Finally, M phase transition is also dysregulated by Tax due to premature activation of the ubiquitin ligase anaphase-promoting complex (APC) and its binding partner cell-division cycle protein 20 (cdc20), which controls the availability of mitotic cyclins for entry into mitosis [208]. Interestingly, centrosome amplification or the induction of supernumerary centrosomes was also observed in Tax-expressing cells through interaction with RanBP1 and re-localization at the centrosome in M phase to trigger aneuploidy [209,210]. 

The ATM (ataxia-telangiectasia mutated) and ATR (ATM- and Rad3-Related) kinases regulate checkpoint activation in response to DNA damage [211]. ATM and ATR are activated upon DNA DSBs or stalled replication forks, respectively, and phosphorylate downstream targets, resulting in p53 activation and cell cycle arrest to allow for the repair of DNA damage or the induction of apoptosis in the event of irreparable damage. Tax inhibits ATM activation and kinase activity following DNA damage due to reduced accumulation of mediator of DNA damage checkpoint 1 (MDC1) at DNA repair foci, thus increasing genomic mutation frequency [212,213]. Base excision repair (BER) corrects small base lesions throughout the cell cycle, and Tax blocks BER triggered by oxidative damage in HTLV-1-transformed T cells [214]. In the presence of a DNA lesion, accumulated p21^waf1/cip1^ interacts with endogenous proliferating cell nuclear antigen (PCNA) to prevent DNA replication while allowing PCNA-dependent DNA repair or nucleotide excision repair (NER). Notably, Tax increases PCNA gene expression to overcome the p21^waf1/cip1^-mediated replication block and to sustain DNA replication by impairing NER, thus promoting genomic instability by elevating the frequency of mutations [215,216,217]. Furthermore, the mismatch repair (MMR) pathway is attenuated in Tax-expressing cells, and mutations of MMR regulatory genes such as hMLH1, hMLH2, hMLH3 and hMLH6 have been observed in ATLL [218]. Double-stranded breaks represent a more severe form of DNA damage that can result in faulty DNA replication and transcription products associated with cellular transformation; these lesions are mainly repaired via homologous recombination (HR). Interestingly, Tax-mediated NF-κB activation results in impaired HR activity that can lead to cellular transformation [219]. 

## 8. Tax Modulation of miRNA Expression for T cell Proliferation and Transformation

MicroRNAs (miRNAs) are a class of small non-coding RNAs that regulate the expression of various genes by the inhibition of protein translation or degradation of target mRNA transcripts upon binding to their untranslated regions (UTRs). The regulation of miRNA expression has garnered much attention due to their crucial regulatory roles in several pathophysiological processes, including cancer [220]. Generally, miRNAs are transcribed as primary miRNAs (pri-miRNAs) by RNA polymerase II followed by processing by the Drosha Ribonuclease III (DROSHA)/DiGeorge syndrome critical region 8 (DGRC8) complex into intermediate precursor miRNAs (pre-miRNAs) within the nucleus [221]. The pre-miRNAs are then transported from the nucleus to the cytoplasm and then cleaved by the endonuclease DICER, which results in the removal of the terminal loop to produce a mature miRNA duplex, while one strand of the duplex protein combines with argonaute (AGO) protein into the RNA-induced silencing complex (RISC) [222].

Several studies have described a dysregulation of miRNA expression in HTLV-1-infected T cell lines, ATLL cell lines, and ATLL patient samples suggesting potential roles in ATLL pathogenesis [111,223,224,225,226,227,228]. Tax interferes with the miRNA machinery by forming a complex with DROSHA to trigger its degradation through the proteasomal pathway, thus repressing its function and resulting in enhanced viral replication [229]. Tax induces the expression of miR-146a and miR-155 in an NF-κB-dependent manner to promote the proliferation of HTLV-1-infected T cells [230,231]. Similarly, miR-130b is also upregulated by Tax in an NF-κB-dependent manner [228]. Tax also downregulates miRNAs including miR-149 and miR-873, possibly to modulate the expression of histone acetyltransferase (HAT) family factors through chromatin remodeling enzymes p/CAF and p300 [232]. Additional miRNAs identified in this study that are downregulated by Tax include miR-135b and miR-872 [232]. A global profiling analysis of miRNAs in ATLL revealed a downregulation of miR-31; interestingly, NIK is an important target of miR-31 [111]. Thus, aberrant expression of NIK and persistent NF-κB activation may be linked to miR-31 repression in ATLL [111]; however, it remains unclear if Tax plays any role in miR-31 suppression. 

## 9. Negative Regulation of Tax through Its Repression by Viral and Cellular genes

Several oncogenic viruses have evolved to counteract host immune signaling by diverse strategies [233]. A key strategy utilized by HTLV-1 is through the tight regulation of the expression of viral proteins to evade host immune surveillance and ensure long-term persistence. Tax is required for viral gene expression and plays important roles in cell transformation by inducing the aberrant expression of anti-apoptotic genes while suppressing the expression of pro-apoptotic genes. Therefore, it is not surprising that Tax is a principal target of CD8+ T cells and antibodies specific for viral antigens [234,235]. However, persistent expression of Tax leads to p21^waf1/cip1^ and p27-mediated cellular senescence through the hyper-activation of NF-κB signaling [208,236]. Tax also interacts with the deubiquitinase USP10 to stimulate ROS production, which can contribute to Tax driven-senescence in HTLV-1-infected T cells [237]. Given that Tax regulates expression from the 5’ LTR, it therefore upregulates structural and enzymatic proteins such as Env, Gag and Pol, which are also targets of immune responses. Since Tax is highly immunogenic, and persistent or high levels of Tax expression induce DNA damage and senescence [236,238], HTLV-1 has evolved unique strategies to tightly control the expression of Tax through its transient but inducible expression by stress stimuli such as hypoxia, p38-MAPK signaling and oxidative stress [239,240,241,242]. It is an emerging theme that Tax and HTLV-1 plus-strand transcription are commonly silenced as a mechanism of immune evasion, but this can be reversed in response to stress stimuli.

Tax is also tightly regulated by other HTLV-1-encoded genes. HTLV-1 p30 inhibits Tax expression by sequestering doubly spliced viral RNA in the nucleus and also blocks Tax *trans*-activation of the viral LTR by competitive binding to CBP/300 [243,244]. Similarly, HBZ also suppresses Tax-mediated transcription and the HTLV-1 LTR by sequestering CREB/ATF transcription factors [245]. However, *HBZ* RNA may increase Tax expression indirectly through the downregulation of p30, thus highlighting the complexity and cross-regulation between different HTLV-1-encoded regulatory genes [246]. Interestingly, constitutively expressed HBZ upregulates the transcription factor BATF3 (basic leucine zipper ATF-like transcription factor 3) as well as BATF3/Interferon regulatory factor 4 (IRF4) target genes by binding to a super-enhancer region in the *BATF3* locus and establishing an auto-regulatory loop; genetic ablation of either BATF3 or IRF4 in ATLL cells induces a profound cell-cycle arrest, possibly due to the downregulation of the target gene *MYC*, even in the absence of Tax [247]. Thus, HBZ plays important roles in the survival of ATLL cells, which commonly exhibit downregulated Tax expression. 

As discussed earlier, Tax protein levels can be downregulated by host E3 ligases such as PDLIM2 which inhibits Tax expression through ubiquitin-mediated degradation in the nuclear matrix [136]. However, PDLIM2 expression is downregulated in HTLV-1-transformed T cells and primary ATLL cells due to DNA methylation that could be reversed by 5-aza-2’deoxycytidine treatment [138]. Constitutive NF-κB signaling is a hallmark of ATLL, even in the absence of Tax, partly due to epigenetic mechanisms such as the downregulation of miR-31 and concomitant upregulation of NIK [111,248]. In addition, activating mutations in critical signaling proteins (i.e., *PLCG1*, *PRKCB* and *CARD11*) in the T-cell receptor (TCR) signaling pathway likely also contribute to persistent NF-κB activation in the absence of Tax [249]. Interestingly, a recent study showed that IRF4 and NF-κB coordinately bind to super enhancers in key growth and survival genes such as *Myc*, *Ccr4* and *Birc3* and form a feed-forward loop driving cell survival in ATLL [250].

Anti-sense RNAs containing the HTLV-1 LTR region, and RNA from the Rex-responsive element in the LTR have been proposed to regulate NF-κB activation by stimulating pattern recognition receptors (PRRs) such as RNA-dependent protein kinase (PKR) and 2’5’-oligoadenylate synthetase, an interferon stimulated gene (ISG), in Tax-negative ATLL cells [251,252]. Interestingly, ATLL patients express high serum levels of the cytokine IL-10, likely induced by NF-κB signaling, which exhibits immunosuppressive functions partly via regulatory T cells (Tregs) in a positive feedback regulation resulting in impairment of Tax-specific CD8+ T cells [253,254,255]. The persistent levels of IL-10 are also maintained through the activation and phosphorylation of signal transducer and activator of transcription 3 (STAT3), which likely plays a role in the expression of anti-apoptotic genes for lymphoproliferation, even in the absence of Tax [256]. Tax expression is downregulated in ~60% of ATLL due to mutations, deletions or epigenetic changes [257]. Intriguingly, Tax expression in a minor fraction of ATLL leukemic cells maintains the whole population of leukemic cells, and Tax expression fluctuates between on and off states [242]. Taken together, HTLV-1 fine-tunes the expression of Tax, which is commonly downregulated in ATLL, to evade anti-viral immune responses and establish a persistent infection which could lead to leukemia/lymphoma in a subset of infected individuals (Figure 3). 

## 10. Conclusions

HTLV-1 has evolved multiple strategies to exploit and disrupt the host cellular machinery for its long-term persistence that can ultimately cause the immortalization and transformation of infected T cells. In this review, we have highlighted the mechanisms of HTLV-1 Tax manipulation and subversion of key cellular pathways and checkpoints and discussed the functional roles of Tax PTMs in the regulation of its protein-protein interactions and subcellular localization. Tax expression is also tightly regulated at the transcriptional and post-translational levels by other HTLV-1 regulatory proteins as well as host factors that optimize viral immune evasion and persistence. Although recent studies have identified stress stimuli and host factors (e.g., p38 and histone H2A) that can regulate Tax expression and the reactivation of latent HTLV-1, there are likely additional pathways and mechanisms important for HTLV-1 latency. Ubiquitination (mono-and poly-) represents a versatile PTM and critical regulatory mechanism for Tax shuttling between cytoplasmic and nuclear compartments as well as chronic NF-κB activation and oncogenicity. Ubiquitination of host proteins is also key for HTLV-1 persistence as histone H2A monoubiquitination plays a critical role in silencing of the HTLV-1 provirus [241]. However, the host factors that regulate Tax PTMs are largely unknown, and thus further studies are needed to identify proteins that modulate Tax ubiquitination, phosphorylation, etc. These host factors could potentially be exploited as drug targets to modulate HTLV-1 latency or antagonize HTLV-1-induced oncogenesis. It also remains unclear how Tax counteracts DUBs to acquire resistance to autophagic or proteasomal degradation despite interacting with several selective autophagy receptors and the proteasome. There are intriguing links between Tax-NF-κB signaling and autophagy, yet the mechanistic basis for Tax utilizing autophagy components such as Beclin1 for NF-κB signaling [258] remains unknown. Finally, it remains poorly understood how NF-κB remains chronically activated in Tax-negative ATLL. Dysregulated microRNAs and somatic mutations in TCR signaling proteins represent two established mechanisms of Tax-independent NF-κB activation, however there are likely additional drivers of NF-κB signaling in ATLL that can emerge in the absence of Tax. 

In addition to Tax, HBZ is a key viral factor important for HTLV-1 persistence and oncogenesis. There exists a complex and cooperative interplay between Tax and HBZ that maintains the fragile equilibrium of NF-κB signaling for driving clonal expansion that is balanced by the inhibition of senescence. This optimal level of signal transduction fine-tuned by Tax-HBZ is likely critical for malignant transformation of infected T cells. Further elucidation of the cross-talk between Tax and HBZ is needed to understand the oncogenic mechanisms of HTLV-1. Another interesting avenue of exploration is to understand how Tax protects and maintains the entire pool of ATLL cells despite its low expression in a negligible fraction of infected cells [242]. Together, research progress in these areas may inform new and efficacious therapeutic interventions to target Tax and/or HBZ in either a pre-ATLL stage or ATLL. 

## Figures and Tables

**Figure 1 pathogens-09-00543-f001:**
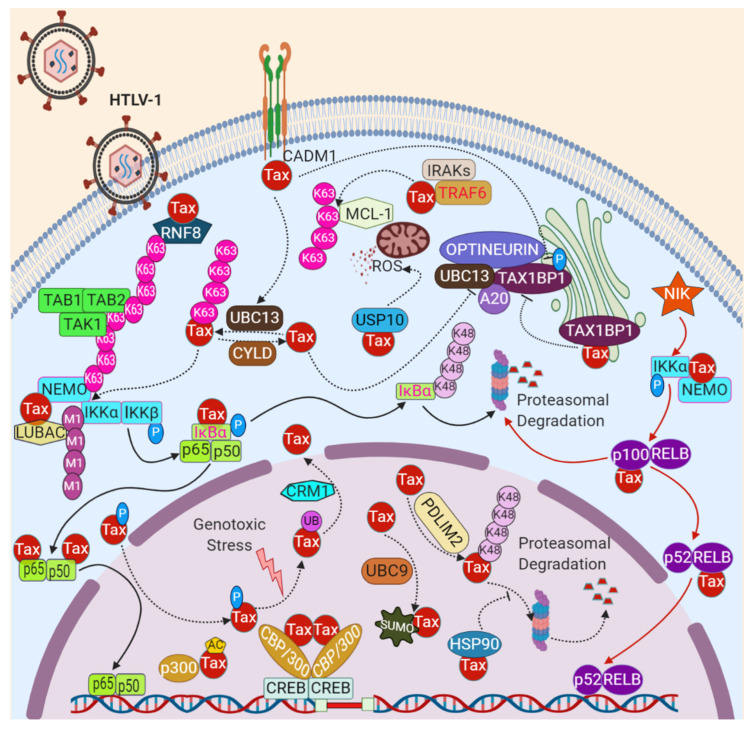
Tax activation of canonical and noncanonical NF-κB signaling. Tax interacts with ring finger protein 8 (RNF8) and linear ubiquitin assembly complex (LUBAC) to recruit K63-linked polyubiquitin chains and linear ubiquitin chains, respectively, to induce IκB kinase (IKK) activation. Tax also interacts with IκBα and p105 to disrupt NF-κB/IκB complexes with subsequent phosphorylation and proteasomal degradation of IκBα and processing to p50, respectively. Tax interacts with and promotes the nuclear translocation of active NF-κB dimers to enhance DNA binding and transcriptional activity. Tax also activates noncanonical NF-κB signaling through interaction with IκB kinase α (IKKα) and NF-κB essential modulator (NEMO) to induce the proteasomal processing of p100 to active p52 and trigger its nuclear translocation along with RelB to activate NF-κB target genes. Tax also interacts with TAX1BP1 to disrupt the Itch, RNF11, NRP and A20 complex to potentiate and sustain NF-κB activation. Cell adhesion molecule 1 (CADM1) interaction with Tax also facilitates Ubc13-mediated K63-linked polyubiquitination of Tax and NF-κB activation. Tax interacts with HSP90 for protection against proteasomal degradation mediated by PDLIM2-triggered K48-linked polyubiquitination in the nuclear matrix. Tax is also monoubiquitinated upon genotoxic stress to trigger its nuclear export in a CRM-1-dependent manner. SUMOylation of Tax is mediated by Ubc9 and SUMOylated Tax is found in nuclear bodies. Acetylated Tax also interacts with p300 in the nucleus. Tax interacts with TRAF6 to stabilize MCL-1 through K63-linked polyubiquitination in mitochondria, and its interaction with USP10 leads to increased reactive oxygen species (ROS) production to inhibit apoptosis. Tax also interacts with CREB-activating transcription factor family protein (CREB/ATF) and recruits CBP/300 to the long terminal repeat (LTR) for *trans*-activation of the viral promoter. The NF-κB canonical pathway is indicated by solid black lines with arrows. The noncanonical NF-κB pathway is indicated by solid red lines with arrows.

**Figure 2 pathogens-09-00543-f002:**
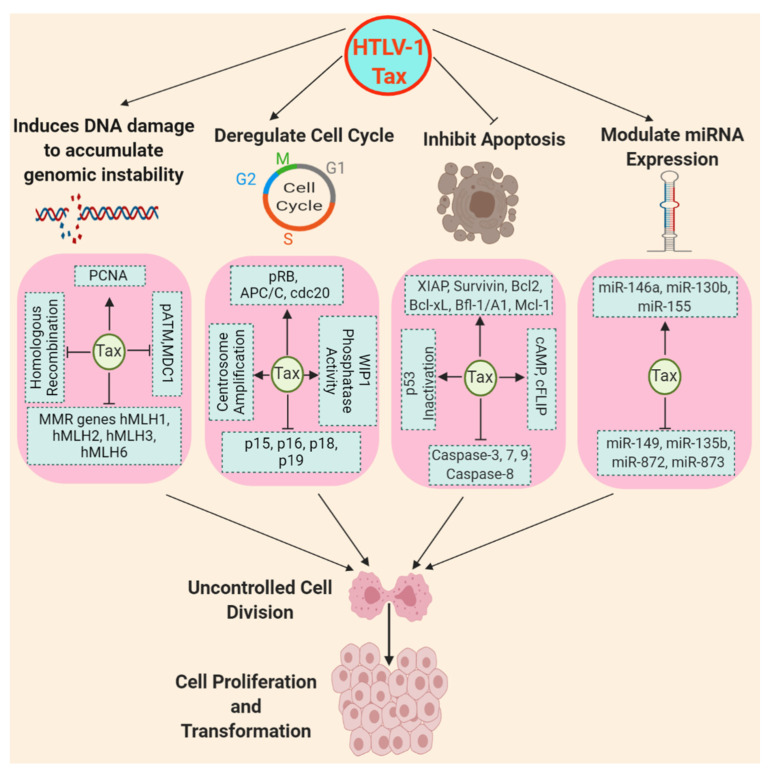
Tax induction of DNA damage, deregulation of the cell cycle, inhibition of apoptosis, and modulation of miRNA expression. Upon DNA damage, Tax suppresses ataxia-telangiectasia mutated (ATM) kinase activation and mediator of DNA damage checkpoint 1 (MDC1) to increase genomic instability with subsequent impairment of homologous recombination, and inhibits mismatch repair (MMR) regulatory genes to block mismatch repair to induce DNA damage. Additionally, Tax increases proliferating cell nuclear antigen (PCNA) expression to overcome the p21^waf1/cip1^-mediated replication block to sustain DNA replication with increased mutation frequency for the accumulation of genomic instability. Tax deregulates the cell cycle and transcribes S phase genes through the hyperphosphorylation of Rb and enhances WIP1 phosphatase activity with concomitant inhibition of CDK4 inhibitors such as p15, p16, p18 and p19, and tumor suppressor p53. Tax also facilitates M phase transition by the activation of anaphase-promoting complex (APC) and its binding partner cdc20 and induces centrosome amplification to trigger aneuploidy. Tax inhibits apoptosis by enhancing the expression of anti-apoptotic proteins such as X-linked inhibitor of apoptosis (XIAP), survivin, BCL-2, Bcl-xL, Bfl-1/A1 and cFLIP and impairs caspase signaling by inactivation of caspase-3, -7, -8 and -9. Tax also stabilizes anti-apoptotic MCL-1 through K63-linked polyubiquitination. Tax promotes functional inactivation of p53 through its phosphorylation and induces expression of Ras proteins and cyclic AMP (cAMP) to augment cell proliferation with increased metabolism. Tax also modulates miRNA expression to fine-tune gene expression for cellular transformation. Tax upregulates miR-146a, miR-130b and miR-155, and downregulates miR-149, miR-135b, miR-872 and miR-873 to promote the proliferation of infected T cells.

**Figure 3 pathogens-09-00543-f003:**
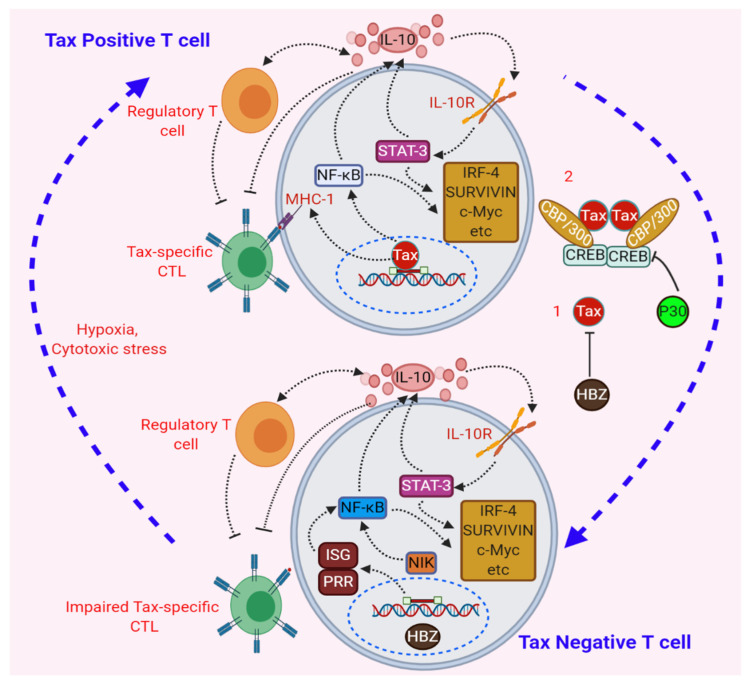
Regulation of Tax expression by viral and cellular mechanisms. The CD8+ T cells specific for Tax antigens are impaired by activated Tregs via the secretion of IL-10 through Tax-mediated NF-κB activation. However, levels of IL-10 are maintained by a positive feedback loop of signal transducer and activator of transcription 3 (STAT3) to maintain the pool of anti-apoptotic genes even in the absence of Tax. As Tax is highly immunogenic, human T-cell lymphotropic virus type 1 (HTLV-1) basic leucine zipper factor (HBZ) suppresses Tax-mediated transcription (1), and HTLV-1 p30 (2) also blocks Tax *trans*-activation of the viral LTR by competitive binding to CBP/300 and sequestering doubly spliced viral RNA in the nucleus. However, Tax expression can be induced by hypoxia and oxidative stress. In Tax-negative cells, HBZ is constitutively expressed, and chronic activation of NF-κB signaling is maintained by aberrant expression of NF-κB-inducing kinase (NIK) to transcribe anti-apoptotic genes for sustained lymphoproliferation. It is also possible that activated interferon-stimulated genes (ISGs)/pattern recognition receptors (PRRs) triggered by sensing of anti-sense RNA containing the HTLV-1 LTR may contribute to NF-κB activation.

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
