# Peer review of "Mechanisms of Oncogenesis by HTLV-1 Tax"

_pathogens, 2020, doi:10.3390/pathogens9070543_

Round 1

Reviewer 1 Report

In this review the authors discuss the role HTLV-1 Tax protein plays in cancer development. The review discusses Tax's interaction with host and cellular proteins to impact signaling pathways, regulation through posttranslational modifications and considers the balancing act between Tax and HBZ.

While the review covers an important topic in the field, there are several major issues which should be addressed prior to publication.

HTLV and the HTLV Tax protein have been studied for nearly 40 years which has resulted in a large body of literature. It is therefore reasonable that not all primary publications can be cited. However, the authors need to be more consistent in how they cite the literature and need to carefully examine their citations to be sure the appropriate papers and/or reviews have been cited. Throughout the review the authors cite incorrect or incomplete lists of references.

For example, in lines 28-30 the authors indicate that two independent groups reported the discovery of HTLV but only cite one of the papers. Both Poiesz et al and Yoshida et al should be cited.  Lines 95-100 discuss viral genomic structure but the references describe protein processing of gag and env. Lines 47-50 identifies the endemic regions for HTLV-1 but only cite references for Australia and Iran. Lines 116 and 117 discuss silencing of the 5'LTR but the reference provided discusses the proliferative response of Tax-1 transducer primary T cells. 

Lines 31-33 state "Although HTLV-1 and HTLV-2 are known to be pathogenic to humans, HTLV-1 is more pathogenic, and the association of HTLV-2 infection with disease has not been clearly established." This seems contradictory. How can HTLV-2 be pathogenic if it is not clearly associated with disease? The authors go on to talk about HTLV-3 and -4 but only cite a paper on disease pathology for HTLV-1 treatment of ATLL.

The authors list various modes of viral entry but only describe the virological synapse (line 81-86; VS). Why? Is it because the VS has been shown to be affected by Tax while the other modes don't have a clear link to Tax? If that is the case this should be explained.

In lines 126-128 the authors discuss Tax 1-4. It is inaccurate to say that only Tax-1 is able to induce disease in transgenic mice and transform murine fibroblasts, etc. Tax-3 and Tax-4 have not been tested and Tax-2 has been shown to immortalize lymphocytes (Ross et al. JVirol 2000 74:2655-62).

The heading for 6 is "HTLV-1-mediated Inhibition of Apoptosis" but the section describes the role Tax plays in not only apoptotic pathways but altering DNA repair, cell cycle and miRNA expression. This heading should be changed to better describe the section. Also in this section the authors should provide a better link/transition to looking at the differences between Tax and HBZ, otherwise it seems out of place.

The authors might also discuss the role Tax plays in altering chromatin structure which in turn alters gene expression. Tax is known to interact with chromatin modifying enzymes such as HDACs (Ego et al 2002; Lu et al 2004), SUV39H1 (Kamoi et al 2006) SMYD# (Yamamoto et al 2011) and EZH2(Fujikawa et al 2016) and Tax-dependent immortalized cells show significant H3K27me3 (a histone modification that indicates repressed chromatin) similar to that of ATL cells.

It would be helpful in Figure 1 to indicate which is the canonical pathway vs the non-canonical pathway.

Figure 3 has numbers in the image but they are not described in the figure legend. 

Author Response

“The authors need to be more consistent in how they cite the literature and need to carefully examine their citations to be sure the appropriate papers and/or reviews have been cited”.  

We thank the reviewer for bringing this issue to our attention and we apologize for the errors in the initial submission in some of the citations. We have carefully examined every citation in the revised manuscript and made appropriate changes to ensure the correct papers have been cited. We have included citations for both Poiesz and Yoshida et al. for the discovery of HTLV-1, and also included all other citations recommended by the reviewer.

“How can HTLV-2 be pathogenic if it is not clearly associated with disease?”

We have deleted the statement in the initial submission regarding the pathogenicity of HTLV-1 and HTLV-2.

“The authors list various modes of viral entry but only describe the virological synapse (line 81-86; VS). Why?”

We have mentioned the different mechanisms of HTLV-1 entry such as the virological synapse, viral biofilms and tunneling nanotubes. However, we did not discuss the other mechanisms of viral entry in detail because they are Tax-independent events and the focus of this review is on Tax mechanisms of oncogenesis.

“In lines 126-128 the authors discuss Tax1-4. It is inaccurate to say that only Tax-1 is able to induce disease in transgenic mice and transform mouse fibroblasts, etc…”

We agree and have deleted this statement in the revised manuscript.

“The heading for 6 is “HTLV-1-mediated inhibition of Apoptosis” but the section described the role Tax plays in not only apoptotic pathways but altering DNA repair, cell cycle and miRNA expression. This heading should be changed to better describe the section”.”

We thank the reviewer for bringing this to our attention. In the revised manuscript, we have separated these sections. Section 6 is focused on HTLV-1 inhibition of apoptosis. Section 7 now focuses on HTLV-1 deregulation of cell cycle and DNA repair. Section 8 is on Tax modulation of miRNAs. Section 9 is on the negative regulation of Tax for immune evasion.

“The authors might also discuss the role Tax plays in altering chromatin structure which in turn regulates gene expression….”

Thank you for this suggestion. We have cited the recommended papers and included more discussion on Tax altering chromatin structure in the revised manuscript.

“It would be helpful in Figure 1 to indicate which is the canonical pathway vs the noncanonical pathway.”

This is a good suggestion. In the revised Figure 1 we have used solid black lines to indicate the canonical NF-kB pathway and solid red lines to indicate the noncanonical pathway. This is described in the Figure 1 legend.

“Figure 3 has numbers in the image but they are not described in the figure legend.”

We apologize for this oversight which has been corrected in the revised manuscript.

Reviewer 2 Report

The article “Mechanisms of Oncogenesis by HTLV-1 Tax” by Mohanty and Harhaj is a well-written and comprehensive review highlighting the several mechanisms used by HTLV-1 Tax to induce cellular transformation and subsequent oncogenesis. The authors focused on Tax oncogenic functions, NFkB activation, post-translational modifications of Tax and their implications to function, inhibition of apoptosis, and the regulation of Tax expression/function. This review is widely useful both to those within the HTLV-1 field and other viral oncology fields. Two minor comments for the authors’ consideration:

  • In Figure 3, there appears to be red numbers 1 and 2 in the figure, but no reference to them in the figure description.
  • Regarding lines 75-76, page 2: HTLV-1 preferentially immortalizes CD4+ T-cells both in cell culture and in vivo, however the virus can infect both CD4 and CD8 T-cells (as well as the other cell types the authors listed). (see reference PMID: 22278223)

Author Response

“In Figure 3, there appears to be red numbers 1 and 2 in the figure, but no reference to them in the figure description.”

This was also brought to our attention by Reviewer 1. We apologize for this oversight which has been corrected in the revised manuscript.

“Regarding lines 75-76, page 2: HTLV-1 preferentially immortalizes CD4+ T cells both in cell culture and in vivo, however the virus can infect both CD4 and CD8 T cells (as well as the other cell types the authors listed).”

We agree with the reviewer and have modified our statement accordingly. The majority of HTLV-1-infected T cells in vivo are CD4+ T cells, and we have cited a paper on this topic. We have also cited additional papers describing other cell types infected by HTLV-1.

Round 2

Reviewer 1 Report

The authors have adequately addressed my concerns.